# Probiotics and Prebiotics Affecting Mental and Gut Health

**DOI:** 10.3390/healthcare12050510

**Published:** 2024-02-21

**Authors:** Merve İnce Palamutoglu, Gizem Köse, Murat Bas

**Affiliations:** 1Department of Nutrition and Dietetics, Institute of Health Sciences, Acibadem Mehmet Ali Aydinlar University, Istanbul 34752, Türkiye; 2Department of Nutrition and Dietetics, Faculty of Health Sciences, Afyonkarahisar Health Sciences University, Afyonkarahisar 03030, Türkiye; 3Department of Nutrition and Dietetics, Faculty of Health Sciences, Acibadem Mehmet Ali Aydinlar University, Istanbul 34752, Türkiye; gizem.kose@acibadem.edu.tr (G.K.); murat.bas@acibadem.edu.tr (M.B.)

**Keywords:** probiotic, prebiotic, gastrointestinal system, depression, anxiety, stress

## Abstract

The effects of the gut microbiota on mental and intestinal health are an area of great interest. This study aimed to reveal the relationship between the intake of probiotic and prebiotic foods and mental and gut health. Data were obtained using an online survey from young adults (n = 538) enrolled at Afyonkarahisar Health Sciences University who agreed to participate in this study in the 2022–2023 academic year. This study included 538 participants, mostly (85.5%) females. Participants who never consumed yogurt had 7.614 times higher Gastrointestinal Symptom Rating Scale scores than those who consumed yogurt daily (*p* < 0.01). Similarly, the frequency of ayran consumption had a statistically significant effect on Bristol Stool Scale scores (*p* < 0.05). The ratio of normal defecation to constipation was 68.7% lower in participants who consumed ayran daily, whereas the ratio of diarrhea to constipation was 76.4% lower in participants who never consumed ayran. However, the frequency of prebiotic consumption did not have a significant effect on Bristol Stool Scale scores (*p* > 0.05). The consumption of probiotic and prebiotic foods exerted a significant effect on GSRS total scores and subfactors of the Depression Anxiety Stress Scale-42, namely depression, anxiety, and stress.

## 1. Introduction

The Food and Agriculture Organization and the World Health Organization have defined probiotic bacteria as live microorganisms that, when consumed in adequate amounts, exert beneficial health effects on the gastrointestinal (GI) tract of the host [1]. Besides regular consumption of probiotic microorganisms in specific amounts, for the host to obtain these benefits, the ingested probiotics must withstand conditions such as stomach acidity, bile salts, and digestive enzymes to reach the intestines in sufficient quantities, colonize the gut, and maintain their viability throughout the product’s shelf life [2]. When probiotic bacteria reach the intestines, they should be at a minimum level of 10^6^–10^8^ colony-forming units (cfu/g) per g or mL [3]. According to several studies, oral doses of >10^9^ cfu per day are required to restore and maintain the balance of bacteria [4]. Lactic acid bacteria, including *Bifidobacterium* and *Lactobacillus* species, and *Saccharomyces boulardii* yeast species are the most important groups of probiotic microorganisms [5]. Probiotic microorganisms (bacteria and yeast cells) which colonize the microbiota can be taken naturally with foods, added to various foods as food supplements, or be ingested in pharmaceutical forms, such as tablets, capsules, powders, sachets, and drops [4]. Probiotics need prebiotics to achieve healthy colonization and sustainability. Prebiotics promote the growth of one or more helpful microorganisms, aiding the probiotic effect and improving the host’s health [6]. Food prebiotics are not digested in the stomach and small intestine. Instead, they undergo fermentation by the microorganisms present in the colon. This fermentation process, which can be beneficial for the host, results in the release of metabolites that serve as a source of energy for these microorganisms. With the intake of prebiotics, the treatment of some patients and prevention of some diseases can be achieved by cultivating a favorable environment within the gut microbiota, thereby promoting a healthy state for the host. Therefore, the regulation of the human intestinal microflora with nutrients has been a popular field in nutritional science [7].

Fructooligosaccharides (FOSs) and galactooligosaccharides (GOSs), along with polyphenols, inulin, and compounds derived from vegetables, herbs, and plants, are the most well-documented prebiotics to treat depression. The administration of FOSs and GOSs in the treatment of gastrointestinal diseases, as well as stress and anxiety-related diseases, indirectly improves host health by promoting the growth of probiotics. Accordingly, prebiotics are often used with probiotics to treat mental disorders such as depression, anxiety, and stress [8].The consumption of foods containing probiotics is one of the therapeutic approaches in functional gastrointestinal diseases, which are chronic and represent a heterogeneous group of diseases that can affect the entire GI tract. These diseases are diagnosed and classified using certain criteria, i.e., not only the duration but also the frequency and quality of defecation gains importance [9]. To distinguish chronic bowel disorders from transient bowel symptoms, according to the Rome IV criteria, a patient’s complaint needs to have started 6 months before admission and continued for 3 months [10]. Irritable bowel syndrome (IBS) is one of the most frequently seen disorders. Individuals with IBS are likely to experience mental health problems such as depression and anxiety owing to the presence of the microbiota, which plays a role in communication with the brain and can affect a person’s mood and behavior [11]. The gut is the natural habitat of an enormous and diverse population of microorganisms that adapts to mucosal surfaces. The symbiotic relationship between the gut bacteria and host is beneficial to both parts, as the host offers a nutrient-rich habitat, whereas the bacteria offer important benefits, such as fermentation that results in the production of short-chain fatty acids, amino acids, and vitamins [12].

The social and personal economic burden of mental diseases is increasing day by day, and depression and anxiety are the two most prevalent mental disorders in terms of personal health expenditures. To reduce the burden of healthcare expenditures, using prebiotics and probiotics may offer possible therapeutic effects on mental illnesses and the healing process [13]. Preclinical studies conducted on mice with dysbiosis have provided compelling evidence, suggesting that probiotic administration can influence behavior and enhance mood. However, whether these findings can be extrapolated to humans is still unclear [14]. Thus, this study aimed to assess the correlation between the consumption of probiotic and prebiotic foods and mental and gut health among young adults.

## 2. Materials and Methods

### 2.1. Study Design

For this study, data were collected through a questionnaire form administered between March and May 2023 to 538 students enrolled at Afyonkarahisar Health Sciences University. In this cross-sectional exploratory study, convenience sampling was applied because we aimed to reach high participation.

### 2.2. Participants, Recruitment, and Sample

In total, 6928 students were enrolled at Afyonkarahisar University of Health Sciences, Faculty of Medicine, Pharmacy, Dentistry and Health Sciences, and Atatürk Health Services Vocational School in 2022–2023 [15]. According to the sample calculation, the sample had to include at least 384 participants with a 95% confidence interval and 5% acceptable error, and a total of 538 students who voluntarily agreed to participate in the study were reached [16]. We excluded participants who were diagnosed with mental diseases and using any kind of medicine or drug.

### 2.3. Data Collection

The questionnaire contained 24 questions and was developed based on a comprehensive literature review conducted before this study. The questionnaire aimed to collect sociodemographic information from the participants [17,18]. The questionnaire was divided into three parts. The first part had 17 questions related to the sociodemographic characteristics of the individuals and their height and body weight. The second part focused on the frequency of consumption of probiotic and prebiotic foods in detail (all cultural and other related foods), and the last part included the scales mentioned below. Online permission for the use of scales in this study was obtained. 

### 2.4. Scales

#### 2.4.1. Gastrointestinal Symptom Rating Scale (GSRS-TR)

Revicki, Wood, Wiklund, and Crawley (1998) developed the GSRS to assess common GI symptoms, clinical experience, and public perceptions of GI symptoms. The 15 items on the GSRS are presented as a 7-point Likert scale, with answers ranging from “no problem” to “severe discomfort”. The GSRS contains five subdimensions based on factor analysis diarrhea, indigestion, constipation, abdominal discomfort, and reflux. In the GSRS, GI-related issues experienced in the previous week were questioned in line with the severity of the problem [19]. The reliability and validity of the Turkish version of the GSRS were assessed by Turan et al. (2017) [20].

#### 2.4.2. Depression Anxiety Stress Scale (DASS-42)

The DASS was developed by Lovibond and Lovibond (1995). This scale is a 42-item tool that measures current (in the past week) symptoms of depression, anxiety, and stress. Each of the three scales consists of 14 items that must be answered on a 0–3 scale (0, the statement “did not apply to me at all”; 3, “it did so frequently or very always)”, and the range of possible scores for each scale is 0–42. In the normal range, scores of 0–9 are used for depression, 0–7 for anxiety, and 0–14 for stress. Scores above these ranges indicate the degree of the problem from mild to extreme [21]. The Turkish version of the DASS-42, including its reliability and validity, was published by Bilgel and Bayram (2010) [22]

#### 2.4.3. Bristol Stool Scale

Blake et al. (2016) validated the Bristol Stool Form Scale in healthy adults and patients with diarrhea-predominant IBS [23]. In the Bristol Stool Scale, questions pertaining to bowel disease characteristics are based on the Rome IV criteria. This section aims to gather information regarding the consistency, thickness, and number of stools [24]. For the analyses, stool consistency was evaluated as a triple categorical variable: constipation (1–2), normal (3–5), and diarrhea (6–7).

#### 2.4.4. Classification of Pro/Prebiotic Foods

Probiotics are living microorganisms that, when provided in sufficient quantities, improve the host’s general health. Previously, dairy products were thought to be the finest preservatives of probiotic bacteria. Owing to increased consumer use, non-dairy probiotic foods have gained greater attention in recent years. Consequently, there is a prospective goal to reduce the effects of non-dairy-based food probiotics, including their matrices from fruits and vegetables, and observations to reduce the effects of dairy-based probiotics are ongoing. Cereals, bread and baked goods, cereal drinks, fruits and vegetables, and, among meats, sucuk are some non-dairy-based probiotic products. Prebiotics are non-digestible food components that benefit the host and give food its textural qualities while promoting the growth and activity of beneficial bacteria in the gut [25]. Breastmilk, soybeans, yams, chicory root, raw oats, unrefined wheat, unrefined barley, non-digestible carbohydrates, and non-digestible oligosaccharides are some sources of prebiotics [26]. In this study, yogurt, kefir, ayran, boza, tarhana, pickles, turnip juice, olives, probiotic yogurt, probiotic milk, and probiotic kefir, which are culturally consumed in Türkiye, were evaluated for their probiotic product consumption frequency. For the evaluation of prebiotic product consumption frequency, whole-grain/mixed-grain bread, oatmeal/breakfast cereals, garlic/onions, tomatoes, Jerusalem artichokes, bananas, cruciferous vegetables (broccoli, cauliflower, radish, etc.), legumes, asparagus, soybeans, nuts (almond and pistachio), and red fruits (rosehip, blueberry, grape, black mulberry, blackberry, etc.) were added to the questionnaire.

### 2.5. Statistical Analysis

Descriptive statistics for categorical variables (demographic characteristics) are presented as frequency and percentage. The conformity of numerical variables to normal distribution was checked using the Shapiro–Wilk test. Descriptive statistics of numerical variables are given as mean ± standard deviation ($\bar{X}$ ± SD) and median (min–max) values for data showing and not showing normal distribution, respectively. The Mann–Whitney U test and Kruskal–Wallis H test were used in the comparison of two independent groups that did not have a normal distribution and of more than two groups, respectively. The results of multiple comparison tests are expressed as letters next to the medians. The relationships between the scales were determined by Spearman’s rank correlation coefficient for data that did not show normal distribution. In the interpretation of correlation coefficients, <0.2 indicates a very weak correlation; 0.2–0.4, weak correlation; 0.4–0.6, moderate correlation; 0.6–0.8, high correlation; and >0.8, very high correlation [27]. Multiple regression analysis and multiple logistic regression analysis were used to test the effect between variables. The level of statistical significance was considered *p* < 0.05, *p* < 0.01, and *p* < 0.001 in all calculations, and interpretations and hypotheses were established as bidirectional. Data analysis was performed in IBM SPSS version 26 (IBM Corp., Armonk, NY, USA).

### 2.6. Ethical Considerations

Ethics approval was obtained from the Afyonkarahisar Health Sciences University Non-Interventional Clinical Research Ethics Committee (dated 3 March 2023 and no. 2023/3). Additionally, consent for this study was acquired from the involved establishments. Verbal and written informed consent was obtained from the participants before this study started.

## 3. Results

A total of 538 young adults were enrolled in this study, and 85.5% (n = 460) were female. The mean age of the participants was 21.38 ± 2.67 years. Among the participants, 68.6% (n = 369) had a normal weight, with a mean body mass index of 22.18 ± 3.62 kg/m^2^. In Table 1, we see that 25.5% (n = 137) of the participants had a chronic disease that was not related to the GI system. Moreover, 79.6% of the participants had undergraduate education, 74.5% had an income that is equal to their expenses, and 66.0% were living at home with their families. In addition, 91.4% of the participants had heard of the term probiotic, and 21.9% used probiotic food supplements daily, of whom 12.3% reported consuming probiotics because of their perceived benefits for the digestive system. However, 56.0% of the participants did not consume probiotics because they did not feel the need for them. In addition, 83.1% of the participants had heard of the term prebiotic, and 53.2% of them thought that probiotics and prebiotics were effective. Again, 75.7% of the participants had normal defecation based on the Bristol Stool Scale, reporting relief after defecation (86.1%), stable defecation frequency (57.1%), and stable defecation status (56.9%) (Table 1). 

Statistically significant differences were found in the scales applied according to the gender of the participants. When the results were examined in female participants, median GSRS total scores (U = 13,893; *p* < 0.01) and the anxiety subfactor score (U = 14,887.5; *p* < 0.05) and stress subfactor score (U = 15,160; *p* < 0.05) of the DASS-42 were statistically higher than the median scores of male participants (Table 2).

When the GSRS total scores (H = 13.479; *p* < 0.01) and depression (H = 9.648; *p* < 0.01), anxiety (H = 12.884; *p* < 0.01), and stress (H =17.134; *p* < 0.001) subfactor scores of the DASS-42 were analyzed according to income status, the median of these scores was statistically higher than the median scores of participants whose income was low compared with their expenses, participants whose expenses were the same as their income, and those with income higher than their expenses (Table 2).

In participants with a chronic disease, median GSRS total scores (U = 18,817.5; *p* < 0.001) and median depression (U =24,172.5; *p* < 0.05), anxiety (U = 20,941.5; *p* < 0.001), and stress (U = 22,122.5; *p* < 0.01) subfactor scores of the DASS-42 were statistically higher than in those without a chronic disease. In the same table, those who were on a special diet program for their disease had statistically higher GSRS total scores (U = 6007; *p* < 0.01) and DASS-42 anxiety (U = 6290; *p* < 0.01) and stress (U = 6398.5; *p* < 0.05) subfactor scores than those who were not (Table 2).

According to the Bristol Stool Scale, the median scores of participants with diarrhea and those with constipation were statistically higher than the median scores of participants with normal defecation (H = 25.616; *p* < 0.001). In the depression subfactor scores of the DASS-42, the median scores of participants with constipation were compared with the median scores of participants who had normal defecation (H = 7.568; *p* < 0.05); in the anxiety subfactor scores, the median scores of participants with constipation and those with diarrhea were compared with the median scores of participants with normal defecation (H = 11.005; *p* < 0.01). The median scores of participants with constipation in the stress subfactor scores were statistically higher than the median scores of participants with normal defecation (H = 7.671; *p* < 0.05). Similar results were obtained in terms of relief after defecation. As shown in Table 2, the median GSRS total scores of participants who experienced relief after defecation were compared with scores of those who did not (U = 14,244; *p* < 0.05). In the anxiety subfactor scores of the DASS-42, the median score of participants who experienced relief after defecation was statistically higher than that of those who did not (U = 14,092; *p* < 0.01).

Participants with and without unstable defecation frequency based on GSRS total scores (U = 23,493; *p* < 0.001) were found to have statistically significantly higher depression subfactor scores (U = 31,123; *p* < 0.05), anxiety subfactor scores (U = 29,288; *p* < 0.01), and stress subfactor scores (U = 29,602.5; *p* < 0.01) of the DASS-42 than those with unstable frequency of all defecation types (Table 2). In the same table, we see that median GSRS total scores (U = 21,829.5; *p* < 0.001) and the depression (U = 31,610; *p* < 0.05), anxiety (U = 28,056.5; *p* < 0.001), and stress (U = 30,478; *p* < 0.01) subfactor scores of the DASS-42 were statistically higher in those who had unstable defecation than in those who did not.

In this study, a significant positive correlation was found between the GSRS total scores of the participants and the depression subfactor scores of the DASS-42 (s = 0.392; *p* < 0.001). Furthermore, significantly and moderately positive correlations were found between GSRS total scores and the anxiety subfactor scores (s = 0.517; *p* < 0.001) and stress subfactor scores (s = 0.489; *p* < 0.001) of the DASS-42 (Table 3).

The participants’ probiotic food consumption frequency was examined and compared to their GSRS total scores. Participants who never consumed yogurt had 7.614 times higher GSRS scores than those who consumed yogurt every day (Table 4). Also, participants who consumed ayran several times a week had 2.732 times lower depression subfactor scores than those who consumed ayran every day, and participants consuming probiotic yogurt several times a week had 6.952 times higher depression subfactor scores than those consuming probiotic yogurt every day. Participants who did not consume any probiotic yogurt had 6.962 times higher depression subfactor scores than those who consumed probiotic yogurt every day, and participants who did not consume probiotic milk had 6.146 times lower depression subfactor scores than those who consumed probiotic milk every day (*p* < 0.05). However, probiotic consumption frequency had no statistically significant effect on the anxiety and stress subfactor scores of the DASS-42.

The participants’ prebiotic food consumption frequency was also examined and compared to their GSRS total scores, and the results revealed that the GSRS total scores of those who never consumed bananas were 3.888 times higher than the scores of those who consumed bananas every day (*p* < 0.05) (Table 4). Moreover, participants who consumed legumes several times a week had 2.798 times lower depression subfactor scores than those who consumed legumes every day. Participants who consumed nuts several times a week had 3.626 times higher depression subfactor scores than those who consumed nuts every day (*p* < 0.05) (Table 4).

When the effect of prebiotics on DASS-42 anxiety subfactor scores was examined, the anxiety subfactor scores of participants who consumed asparagus several times a week were found to be 8.166 times lower than the scores of those who consumed asparagus every day. Moreover, participants who consumed soybeans several times a week had 7.955 times higher anxiety subfactor scores than those who consumed soybeans every day, and the anxiety subfactor scores of participants who did not consume any soybeans were 7.379 times higher than the scores of those who consumed soybeans every day. In addition, the anxiety subfactor scores of participants who consumed nuts several times a week were 2.270 times higher than the scores of those who consumed nuts every day (*p* < 0.05) (Table 4). The effects of prebiotic food consumption frequency on DASS-42 stress subfactor scores were significant, as participants who never consumed soybeans had 7.958 times higher stress subfactor scores than those who consumed soybeans every day. In addition, participants who consumed nuts several times a week had 3.727 times higher stress subfactor scores than those who consumed nuts every day (*p* < 0.05) (Table 4).

The association between the participants’ probiotic consumption frequency and the Bristol Stool Scale was evaluated. Ayran consumption frequency was found to exert a statistically significant effect on the Bristol Stool Scale scores (*p* < 0.05). Accordingly, the ratio of normal defecation to constipation was 68.7% lower in young adults who consumed ayran every day. In addition, the ratio of diarrhea to constipation was 76.4% lower in those who consumed ayran every day. The frequency of prebiotic consumption did not affect the Bristol Stool Scale scores of the study participants (*p* > 0.05) (Table 5).

## 4. Discussion

This study aimed to reveal the relationships between probiotic and prebiotic food consumption, stress, anxiety, and depression, and gut health. The consumption of some probiotic and prebiotic foods positively affects stress, depression, anxiety, and gut health [13]. In this study, most of the young adult participants stated that they had heard the terms probiotic and prebiotic, and more than half of the participants stated that they were beneficial. In another study, Allisa (2021) stated that 56.0% of participants used probiotic/prebiotic supplements, 44.3% preferred milk and dairy products as a probiotic/prebiotic food source, and most of them stated that probiotics and prebiotics were beneficial for digestion. In addition, a high correlation was found between education level and knowledge of probiotics and prebiotics [28]. The participants preferred probiotics and prebiotics primarily because of their positive effects on health. Regarding income status, GSRS and DASS-42 scores were higher in participants whose income was lower than their expenses.

In this study, the frequency of consumption of probiotic products such as ayran, probiotic yogurt, and probiotic milk had a statistically significant effect on DASS-42 depression subfactor scores. In this study, depression subfactor scores were higher in participants who consumed probiotic yogurt more than several times a week and those who never consumed it compared to participants who consumed it every day, which may indicate that regular probiotic yogurt consumption does not have any beneficial effect on mental health. The fact that participants who did not consume probiotic milk had lower depression subfactor scores than those who consume probiotic milk every day may indicate that probiotic milk consumption has a beneficial effect on mental health. Since there is still no convincing evidence about the effect of probiotics on mental health, the negative effect of probiotic yogurt suggests that these results may have been caused by something other than probiotics. Participants who consumed ayran several times a week had lower depression scores, but those who consumed probiotic yogurt several times a week had higher depression scores than those consuming these foods every day. Similar results were found for probiotic milk and yogurt, too. These results can be caused by the amount of consumption. Chahwan et al. (2019) analyzed 71 participants with depressive symptoms using the DASS-21 and Beck Depression Index-II (BDI) and evaluated the differences in the microbiota of these individuals after 8 weeks of probiotic/placebo treatment. On the BDI scale, the post-intervention probiotic group was found to have lower depressive symptoms than the placebo group. In the probiotic group, participants with milder depression had lower dysfunctional attitude scores. By contrast, no effects were observed in people with severe depression and those taking a placebo [29]. In another study, Venkataraman et al. (2020) analyzed 74 participants preparing for exams using the Perceived Stress Scale, DASS, and State-Trait Anxiety Inventory questionnaires and found improvement in psychological symptoms in the probiotic group compared with the placebo group [30]. In their pilot study, Siegel and Conklin (2020) examined the role of probiotics in reducing stress, anxiety, and depression in 40 participants after a week of supplementation. The results showed no significant difference between the probiotic and placebo groups. Overall, 1 week of supplementation did not reduce stress, depressive symptoms, or anxiety in healthy young adults [31]. Yu et al. (2018) reported that higher levels of yogurt consumption (1–3 times/week, 4–7 times/week, and 2 times/day) were associated with enhanced depression symptoms by ratios of 1.05 (0.96, 1.15), 1.02 (0.91, 1.15), and 2.10 (1.61, 2.73) compared with the lowest consumption group (1 time/week or almost never) [32]. In a meta-analysis, Sikorska et al. (2023) examined the molecular mechanisms underlying the use of probiotics to treat anxiety and depression symptoms in healthy people and depressed patients with or without somatic disorders. Probiotic administration was shown to have a positive effect on patients with depression, particularly in those with concomitant chronic diseases. A higher efficacy was observed in patients with depression and somatic disorders than in those with depression alone. The differences in the effect of probiotic usage between studies was said to be attributed to factors such as timing of administration, bacterial strain diversity, and usage of probiotics or symbiotics. The review underscored the potential benefits of probiotics in managing depression and anxiety symptoms, particularly in individuals with concomitant somatic disorders. Also, complex interactions between the gut microbiota, immune response, and brain function were highlighted, providing information about prospective treatment options for mental health problems [33]. The effects of probiotics on the microbiota–gut–brain axis show potential as a complementary treatment for some psychiatric disorders, particularly depression. Moreover, more studies are needed to better understand their effects, mechanisms, and optimal applications in mental health treatment [34]. While similar studies found that probiotic supplementation had no effect or even had the opposite effect, some studies have found that the frequency of probiotic use had a statistically significant influence on depression.

In this study, the frequency of consumption of bananas, which are a prebiotic food, had a statistically significant effect on the total GSRS scores of the participants. Moreover, the frequency of the consumption of legumes and nuts as prebiotics had a positive effect on depression. The frequency of consumption of prebiotic asparagus, soybeans, and nuts showed a statistically significant effect on anxiety. In addition, those who consumed prebiotic nuts and legumes several times a week or never had higher depression, anxiety, and stress subfactor scores than those who consumed these foods daily. Nut and legume consumption might not have any effect on depression, anxiety, and stress, and these results may depend on amounts consumed. Karbownik et al. (2022) evaluated the cognitive performance and depression and anxiety symptoms of medical school students who consumed fermented foods and food-derived prebiotics for 7 days before an exam. The 1-week consumption of each fermented and prebiotic-containing food item was tested separately for its association with cognitive performance under stress, and the results revealed that none of the products were associated with cognition. On the contrary, a high consumption of asparagus, chicory roots, dandelion leaves, globe artichokes, and Jerusalem artichokes was associated with anxiety severity [35]. In a rat study, it was found that bananas can have a potential therapeutic effect on diabetes-related mood disorder treatment via increased plasma serotonin levels and modulation of the gut–microbiota–brain axis [36]. Prebiotic food intake may have a positive effect on stress. Also, in a systematic review, it was stated that even if there is very limited evidence of positive outcomes of prebiotics and probiotics in psychiatric disorders, they seem to be beneficial for major depressive disorder and Alzheimer’s disease [37].

The consumption frequency of probiotics such as ayran exerted a statistically significant effect on the Bristol Stool Scale, but the frequency of prebiotic consumption did not affect the Bristol Stool Scale. In a human study, Moreira et al. (2017) divided 49 patients diagnosed with irritable bowel disease into two groups using the Bristol Stool Scale. One group was given a probiotic cultured milk drink, and the other group was given a probiotic non-cultured milk drink. Considering the Bristol Stool Scale, improvements were detected in the shape and consistency of stools in both groups; however, no statistically significant difference was observed between the groups [10]. Gotteland et al. (2010) evaluated the effect of consuming probiotic-containing dairy products and prebiotics on the digestive comfort of healthy individuals and those with constipation. The results were more frequent bowel movements and a softer stool consistency [38]. When the trials were analyzed, how probiotic and prebiotic food consumption affected stool frequency and consistency was unclear. Thus, more research is required. In a comprehensive analysis of intervention trials, it was observed that probiotic and symbiotic supplements showed a modest improvement in reducing depression symptoms compared to placebo, but prebiotic supplementation did not demonstrate the same benefit [39]. The clinician guidelines for the treatment of psychiatric disorders with nutraceuticals and phytochemicals state that microbial dysbiosis may be more common in older people and lead to increased oxidative and inflammatory load (including on the digestive system). In the future, studies in these areas will help inform more individualized interventions, particularly those involving prebiotic or probiotic products [40].

In the past decade, interest in how the gut microbiota affect mental and cognitive health in various ways has increased. The gut microbiota have been linked to memory, learning, anxiety, stress, and brain disorders, and ways of using microbiota-targeting nutrition and treatments for brain health have been explored. However, gaps remain in understanding these mechanisms, with most of the evidence coming from animal studies rather than clinical studies. Standardized methodologies, advanced models, omics technologies, and interdisciplinary collaborations for human studies are needed to fill these gaps [41].


*Strengths, Limitations, and Future Research*


The strengths of this study include the use of face-to-face interviews and the reliability of the results. However, this study had some limitations. First, the study population was students, and secondly, because of the sample, the mean age was low (young adults). Also, the lack of information on the participants’ living environments was a limitation, as we did not control for stressful environments as a risk factor for depression or anxiety. Thirdly, this study did not present data on gut microbiota sequencing. Consequently, this investigation did not examine the effects of prebiotic and probiotic food consumption on the gut microbiome community. Further studies are needed to evaluate the factors that affect individuals’ use of probiotics/prebiotics and the contribution of probiotic strains to mental and gut health.

## 5. Conclusions

The gut microbiota are associated with psychological phenomena such as stress and anxiety by communicating with the brain in two ways. This study provides new information on the relationship of probiotics/prebiotics with mental and intestinal health.

## Figures and Tables

**Table 1 healthcare-12-00510-t001:** Sociodemographic and health-related variables of the participants (n = 538).

	n	%
**Gender**		
Male	78	14.5
Female	460	85.5
**Age** ($\bar{X}\pm SS$)	21.38 ± 2.67	
**BMI Group**		
Underweight	71	13.2
Normal	369	68.6
Pre-obese	98	18.2
**BMI** ($\bar{X}\pm SS$)	22.18 ± 3.62	
**Educational level**		
Associate degree	110	20.4
Bachelor’s degree	428	79.6
**Level of income**		
Income lower than expenses	88	16.4
Expenses same as income	401	74.5
Income higher than expenses	49	9.1
**Household**		
At home with family	355	66.0
At home alone	30	5.6
At home with friends	29	5.4
At dorm alone	16	3.0
At dorm with friends	104	19.3
At home with relatives	4	0.7
**Diagnosed diseases (other than GI)**		
Have a diagnosed disease	137	25.5
Not diagnosed with any disease	401	74.5
**Special diet for disease**		
Follow a diet	34	6.3
Do not follow a diet	504	93.7
**Probiotic knowledge (self-assessment)**		
Know of probiotics	492	91.4
Do not know of probiotics	46	8.6
**Probiotic supplement usage**		
Using probiotic supplement	118	21.9
Not using probiotic supplement	420	78.1
**Reason for usage of probiotics**		
I don’t consume	420	78.1
Beneficial for the digestive system	66	12.3
I think it protects against cancer	1	0.2
I find it delicious	14	2.6
It strengthens the immune system	37	6.9
**Why I don’t use probiotics**		
I don’t know what it is	98	23.3
I do not find it natural	19	4.5
I do not need	235	56.0
I find it expensive	53	12.6
I do not trust the content	15	3.6
**Prebiotic knowledge (self-assessment)**		
Know of prebiotics	447	83.1
Do not know of prebiotics	91	16.9
**Thinking probiotics and prebiotics have an effect**		
They have an effect	286	53.2
They partially have an effect	228	42.4
They do not have an effect	24	4.5
**Bristol Stool Scale**		
Constipation	57	10.5
Normal	407	75.7
Diarrhea	74	13.8
**Relief after defecation**		
Feel relief after defecation	463	86.1
Do not feel relief after defecation	75	13.9
**Unstable frequency of defecation**		
Unstable frequency of defecation	231	42.9
Stable frequency of defecation	307	57.1
**Unstable defecation status**		
Unstable defecations	232	43.1
Stable defecations	306	56.9

**Table 2 healthcare-12-00510-t002:** Comparison of GSRS and DASS-42 scores according to demographic and health characteristics, probiotic and prebiotic knowledge, and defecation.

	GSRS Total	Depression	Anxiety	Stress
	Median (Min–Max)	Median (Min–Max)	Median (Min–Max)	Median (Min–Max)
**Gender**				
Male	21.5 (15–71)	12 (0–42)	5.5 (0–34)	12 (0–37)
Female	27 (15–90)	13 (0–42)	8 (0–42)	15 (0–42)
**U**	13,893	16,980.5	14,887.5	15,160
** *p* **	0.001 **	0.449	0.016 *	0.028 *
**Level of income**				
Income lower than expenses	33 ^b^ (15–77)	14 ^b^ (0–42)	11 ^b^ (0–42)	19 ^b^ (0–42)
Expenses same as income	25 ^a^ (15–90)	12 ^a^ (0–42)	8 ^a^ (0–37)	14 ^a^ (0–42)
Income higher than expenses	24 ^a^ (15–62)	12 ^a^ (0–42)	6 ^a^ (0–25)	14 ^a^ (0–35)
**H**	13.479	9.648	12.884	17.134
** *p* **	0.001 **	0.008 **	0.002 **	<0.001 ***
**Diagnosed diseases (other than GI)**				
Have a diagnosed disease	32 (15–90)	15 (0–42)	11 (0–42)	18 (0–42)
Not diagnosed with any disease	24 (15–90)	12 (0–42)	7 (0–40)	14 (0–42)
**U**	18,817.5	24,172.5	20,941.5	22,122.5
** *p* **	<0.001 ***	0.036 *	<0.001 ***	0.001 **
**Special diet for disease**				
Follow a diet	35 (15–73)	14 (0–42)	12.5 (1–39)	20 (0–41)
Do not follow a diet	25 (15–90)	13 (0–42)	8 (0–42)	14 (0–42)
**U**	6007	7790.5	6290	6398.5
** *p* **	0.003 **	0.375	0.009 **	0.013 *
**Probiotic knowledge (self-assessment)**				
Know of probiotics	26 (15–90)	13 (0–42)	8 (0–42)	15 (0–42)
Do not know of probiotics	24 (15–90)	13 (0–42)	8 (0–37)	14 (0–38)
**U**	10,506	11,228.5	10,989.5	10,368.5
** *p* **	0.421	0.931	0.746	0.347
**Probiotic usage**				
Using probiotics	26.5 (15–90)	13 (0–42)	9.5 (0–36)	16 (0–40)
Not using probiotics	26 (15–90)	13 (0–42)	8 (0–42)	14 (0–42)
**U**	23,358	24,214	22,529.5	22,599.5
** *p* **	0.340	0.704	0.131	0.144
**Prebiotic knowledge (self-assessment)**				
Know of prebiotics	26 (15–90)	13 (0–42)	8 (0–42)	15 (0–42)
Do not know of prebiotics	25 (15–66)	13 (0–42)	7 (0–37)	13 (0–42)
**U**	19,559	19,862.5	19,110	18,899.5
** *p* **	0.564	0.725	0.363	0.287
**Bristol Stool Scale**				
Constipation	31 ^b^ (15–73)	15 ^b^ (0–42)	11 ^b^ (0–42)	17 ^b^ (0–41)
Normal	25 ^a^ (15–77)	12 ^a^ (0–42)	7 ^a^ (0–40)	14 ^a^ (0–42)
Diarrhea	32 ^b^ (16–90)	14 ^ab^ (0–41)	10 ^b^ (0–34)	16 ^ab^ (0–40)
**H**	25.616	7.568	11.005	7.671
** *p* **	<0.001 ***	0.023 *	0.004 **	0.022 *
**Relief after defecation**				
Feel relief after defecation	27 (15–90)	13 (0–42)	8 (0–40)	15 (0–42)
Do not feel relief after defecation	23 (15–73)	12 (0–42)	5 (0–42)	13 (0–41)
**U**	14,244	16,007	14,092	15,190
** *p* **	0.012 *	0.277	0.009 **	0.082
**Unstable frequency of defecation**				
Unstable frequency of defecation	31 (15–90)	13 (0–42)	10 (0–42)	16 (0–42)
Stable frequency of defecation	23 (15–73)	12 (0–42)	7 (0–36)	13 (0–42)
**U**	23,493	31,123	29,288	29,602.5
** *p* **	<0.001 ***	0.015 *	0.001 **	0.001 **
**Unstable defecation status**				
Unstable defecations	32 (15–90)	14 (0–42)	10 (0–42)	16 (0–42)
Stable defecations	22,5 (15–90)	12 (0–42)	6 (0–39)	14 (0–42)
**U**	21,829.5	31,610	28,056.5	30,478
** *p* **	<0.001 ***	0.029	<0.001 ***	0.005 **

GSRS: Gastrointestinal Symptom Rating Scale; U: Mann–Whitney U Test; H: Kruskal–Wallis H Test; s: Spearman rank differences correlation coefficient; * *p* < 0.05; ** *p* < 0.01; *** *p* < 0.001; ^a,b^: Difference between medians that do not have the same letter is significant (*p* < 0.05).

**Table 3 healthcare-12-00510-t003:** Correlation coefficients between participants’ GSRS total scores and DASS-42 subfactor scores.

	GSRS Total	*p*
**Depression**	0.392	<0.001 ***
**Anxiety**	0.517	<0.001 ***
**Stress**	0.489	<0.001 ***

GSRS: Gastrointestinal Symptom Rating Scale; s: Spearman rank differences correlation coefficient; *** *p* < 0.001.

**Table 4 healthcare-12-00510-t004:** The effect of participants’ probiotic and prebiotic consumption frequency on GSRS total scores and DASS-42 subfactor scores.

	GSRS			DASS-42			Anxiety			Stress		
	β	t	*p*	β	t	*p*	β	t	*p*	β	t	*p*
**Probiotics**												
**(Constant/everyday consumption)**	42.090	6.904	<0.001 ***	19.487	3.970	<0.001 ***	21.569	5.744	<0.001 ***	22.275	4.976	<0.001 ***
**Yogurt (Ref: Daily)**												
Several Times a Week	1.088	0.810	0.418	−0.026	−0.024	0.981	−0.730	−0.883	0.378	−0.562	−0.570	0.569
Never	7.614	2.815	0.005 **	0.844	0.387	0.699	0.438	0.263	0.793	1.321	0.665	0.506
**Kefir (Ref: Daily)**												
Several Times a Week	−8.320	−1.798	0.073	−1.433	−0.384	0.701	−4.169	−1.463	0.144	−5.120	−1.507	0.132
Never	−8.172	−1.838	0.067	−0.840	−0.235	0.814	−3.218	−1.175	0.240	−4.390	−1.345	0.179
**Ayran (Ref: Daily)**												
Several Times a Week	−0.783	−0.477	0.633	−2.732	−2.069	0.039 *	−0.643	−0.637	0.524	−0.743	−0.617	0.537
Never	0.135	0.068	0.946	0.405	0.252	0.801	0.261	0.212	0.832	0.804	0.549	0.583
**Boza (Ref: Daily)**												
Several Times a Week	−1.521	−0.172	0.864	−9.045	−1.269	0.205	−7.701	−1.412	0.158	−4.424	−0.681	0.496
Never	2.097	0.293	0.770	−3.359	−0.583	0.560	−4.315	−0.979	0.328	−0.053	−0.010	0.992
**Tarhana (Ref: Daily)**												
Several Times a Week	−1.350	−0.590	0.555	0.046	0.025	0.980	−2.049	−1.455	0.146	−1.226	−0.730	0.466
Never	0.951	0.429	0.668	2.112	1.184	0.237	−0.134	−0.098	0.922	0.409	0.251	0.802
**Pickles (Ref: Daily)**												
Several Times a Week	0.797	0.511	0.610	−0.708	−0.564	0.573	−0.314	−0.326	0.744	−0.869	−0.759	0.448
Never	−0.453	−0.252	0.801	−0.933	−0.645	0.519	0.433	0.392	0.696	−0.448	−0.340	0.734
**Turnip Juice (Ref: Daily)**												
Several Times a Week	0.465	0.112	0.911	−0.329	−0.099	0.921	−1.039	−0.408	0.684	0.323	0.106	0.915
Never	−1.842	−0.496	0.620	−4.141	−1.386	0.166	−4.045	−1.771	0.077	−3.288	−1.207	0.228
**Olives (Ref: Daily)**												
Several Times a Week	0.186	0.113	0.910	1.836	1.376	0.169	−0.187	−0.183	0.855	−0.117	−0.096	0.923
Never	−1.091	−0.588	0.557	−0.019	−0.013	0.990	−0.059	−0.052	0.959	0.118	0.087	0.931
**Probiotic Yogurt (Ref: Daily)**												
Several Times a Week	1.115	0.301	0.763	6.952	2.331	0.020 *	2.431	1.066	0.287	3.913	1.439	0.151
Never	−0.656	−0.202	0.840	6.962	2.666	0.008 **	2.711	1.357	0.175	3.595	1.510	0.132
**Probiotic Milk (Ref: Daily)**												
Several Times a Week	0.057	0.013	0.989	−6.018	−1.726	0.085	−0.877	−0.329	0.742	−2.873	−0.904	0.367
Never	−2.398	−0.621	0.535	−6.146	−1.976	0.049 *	−2.187	−0.919	0.358	−4.700	−1.657	0.098
**Probiotic Kefir (Ref: Daily)**												
Several Times a Week	−6.334	−1.427	0.154	2.356	0.659	0.510	0.106	0.039	0.969	1.216	0.373	0.709
Never	−2.360	−0.625	0.532	3.099	1.020	0.308	0.641	0.276	0.783	3.467	1.251	0.211
**Prebiotics**												
**(Constant)**	26.819	5.547	<0.001 ***	11.166	2.823	0.005 **	11.632	3.886	<0.001 ***	14.742	4.127	<0.001 ***
**Whole-Grain/Mixed-Grain Breads (Ref: Daily)**												
Several Times a Week	0.443	0.252	0.801	0.647	0.451	0.652	0.240	0.221	0.825	0.407	0.314	0.753
Never	−0.797	−0.557	0.578	−0.250	−0.213	0.831	0.155	0.175	0.861	−0.236	−0.224	0.823
**Oatmeal/Breakfast Cereals (Ref: Daily)**												
Several Times a Week	−2.085	−0.968	0.334	−0.279	−0.158	0.874	−2.119	−1.589	0.113	−2.216	−1.392	0.165
Never	−2.304	−1.168	0.243	−0.870	−0.539	0.590	−2.274	−1.862	0.063	−1.794	−1.231	0.219
**Garlic/Onions (Ref: Daily)**												
Several Times a Week	−1.304	−0.892	0.373	0.374	0.313	0.755	−0.272	−0.301	0.764	−0.380	−0.352	0.725
Never	−1.125	−0.591	0.555	1.766	1.134	0.257	0.534	0.453	0.651	−0.155	−0.110	0.912
**Tomatoes (Ref: Daily)**												
Several Times a Week	2.172	1.463	0.144	1.002	0.825	0.410	1.153	1.254	0.210	0.898	0.819	0.413
Never	−3.363	−1.229	0.220	0.692	0.309	0.757	0.775	0.457	0.648	0.443	0.219	0.827
**Jerusalem artichokes (Ref: Daily)**												
Several Times a Week	4.814	1.501	0.134	−0.362	−0.138	0.890	1.701	0.857	0.392	−0.619	−0.261	0.794
Never	−0.405	−0.151	0.880	1.165	0.531	0.595	1.186	0.715	0.475	0.237	0.120	0.905
**Bananas (Ref: Daily)**												
Several Times a Week	−0.398	−0.253	0.801	−1.002	−0.779	0.436	−1.518	−1.559	0.120	−1.146	−0.986	0.324
Never	3.888	1.996	0.046 *	−0.305	−0.192	0.848	0.783	0.649	0.517	0.723	0.503	0.615
**Cruciferous Veg. (Ref: Daily)**												
Several Times a Week	−1.818	−0.903	0.367	0.076	0.046	0.963	−0.872	−0.699	0.485	−0.436	−0.293	0.769
Never	1.255	0.578	0.563	0.528	0.297	0.766	0.083	0.062	0.951	0.522	0.325	0.745
**Legumes (Ref: Daily)**												
Several Times a Week	−0.186	−0.110	0.912	−2.798	−2.027	0.043 *	−1.018	−0.975	0.330	−1.175	−0.943	0.346
Never	0.694	0.319	0.750	−1.809	−1.017	0.309	−0.258	−0.192	0.848	−0.568	−0.354	0.724
**Asparagus (Ref: Daily)**												
Several Times a Week	1.070	0.164	0.870	−5.104	−0.954	0.341	−8.166	−2.016	0.044 *	−5.088	−1.053	0.293
Never	−1.136	−0.207	0.836	−3.161	−0.703	0.482	−6.644	−1.953	0.051	−3.754	−0.925	0.355
**Soybeans (Ref: Daily)**												
Several Times a Week	1.972	0.366	0.714	5.252	1.192	0.234	7.955	2.386	0.017 *	7.112	1.788	0.074
Never	6.854	1.541	0.124	6.943	1.908	0.057	7.379	2.679	0.008 **	7.958	2.421	0.016 *
**Nuts (Ref: Daily)**												
Several Times a Week	1.108	0.626	0.531	3.626	2.505	0.013 *	2.270	2.072	0.039 *	3.727	2.852	0.005 **
Never	2.066	1.082	0.280	2.657	1.701	0.090	1.593	1.347	0.178	2.522	1.788	0.074

β: Beta coefficient. t: Independent sample *t*-test. * *p* < 0.05; ** *p* < 0.01; *** *p* < 0.001.

**Table 5 healthcare-12-00510-t005:** The effect of participants’ probiotic and prebiotic consumption frequency on Bristol Stool Scale scores.

					OR (95% Cl)	
		OR	Wald	*p*	Lower Limit	Upper Limit
Normal	(Constant)	1.956	2.383	0.123	-	-
	**Yogurt (Ref: Daily)**					
	Several Times a Week	1.066	0.040	0.842	0.568	2.001
	Never	1.244	0.126	0.723	0.372	4.163
	**Kefir (Ref: Daily)**					
	Several Times a Week	0.381	0.609	0.435	0.034	4.307
	Never	0.786	0.039	0.844	0.071	8.654
	**Ayran (Ref: Daily)**					
	Several Times a Week	0.537	1.823	0.177	0.218	1.324
	Never	0.313	5.268	0.022 *	0.116	0.844
	**Tarhana (Ref: Daily)**					
	Several Times a Week	1.983	1.452	0.228	0.651	6.041
	Never	0.802	0.187	0.666	0.294	2.184
	**Pickles (Ref: Daily)**					
	Several Times a Week	0.656	1.131	0.287	0.302	1.427
	Never	0.689	0.719	0.396	0.291	1.630
	**Turnip Juice (Ref: Daily)**					
	Several Times a Week	0.926	0.006	0.938	0.134	6.414
	Never	1.085	0.008	0.927	0.188	6.260
	**Olives (Ref: Daily)**					
	Several Times a Week	1.121	0.087	0.769	0.523	2.404
	Never	1.024	0.003	0.958	0.427	2.451
	**Probiotic Yogurt (Ref: Daily)**					
	Several Times a Week	1.325	0.110	0.740	0.251	6.996
	Never	1.050	0.004	0.948	0.240	4.599
	**Probiotic Milk (Ref: Daily)**					
	Several Times a Week	0.718	0.129	0.720	0.117	4.400
	Never	1.793	0.451	0.502	0.326	9.853
	**Probiotic Kefir (Ref: Daily)**					
	Several Times a Week	3.301	1.216	0.270	0.395	27.577
	Never	1.711	0.418	0.518	0.336	8.712
Diarrhea	(Constant)	0.459	0.098	0.754		
	**Yogurt (Ref: Daily)**					
	Several Times a Week	1.287	0.397	0.529	0.588	2.817
	Never	2.011	0.871	0.351	0.464	8.719
	**Kefir (Ref: Daily)**					
	Several Times a Week	0.224	1.170	0.279	0.015	3.374
	Never	0.530	0.218	0.641	0.037	7.618
	**Ayran (Ref: Daily)**					
	Several Times a Week	0.455	2.131	0.144	0.158	1.309
	Never	0.236	5.454	0.020 *	0.070	0.793
	**Tarhana (Ref: Daily)**					
	Several Times a Week	2.179	1.221	0.269	0.547	8.679
	Never	1.095	0.019	0.889	0.305	3.929
	**Pickles (Ref: Daily)**					
	Several Times a Week	0.868	0.084	0.772	0.334	2.255
	Never	0.624	0.730	0.393	0.212	1.841
	**Turnip Juice (Ref: Daily)**					
	Several Times a Week	0.727	0.070	0.792	0.068	7.729
	Never	0.827	0.030	0.862	0.097	7.044
	**Olives (Ref: Daily)**					
	Several Times a Week	1.199	0.142	0.707	0.466	3.089
	Never	1.440	0.456	0.499	0.500	4.146
	**Probiotic Yogurt (Ref: Daily)**					
	Several Times a Week	1.393	0.104	0.747	0.185	10.465
	Never	0.753	0.094	0.759	0.122	4.638
	**Probiotic Milk (Ref: Daily)**					
	Several Times a Week	2.334	0.474	0.491	0.209	26.053
	Never	5.112	1.954	0.162	0.519	50.363
	**Probiotic Kefir (Ref: Daily)**					
	Several Times a Week	2.010	0.300	0.584	0.165	24.427
	Never	0.880	0.015	0.901	0.116	6.644

95% confidence interval and *p* values. OR: odds ratio. * *p* < 0.05.

## Data Availability

The original contributions presented in this study are included in the article. Further inquiries can be directed to the corresponding author.

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
