# Peer review of "Probiotics and Prebiotics Affecting Mental and Gut Health"

_healthcare, 2024, doi:10.3390/healthcare12050510_

Round 1
Reviewer 1 Report
Comments and Suggestions for Authors
The manuscript by Palamutoglu et al. is well-written. However, the manuscript needs to address the major comments to better describe the results in the discussion section.
Major comments:
1. Why do participants consuming probiotic yogurt several times a week or never have higher depression subfactor scores than those consuming daily? This needs to be addressed in the discussion section.
2. Why do participants who never consume probiotic milk have lower depression subfactor scores than those who consume daily? This needs to be addressed in the discussion section.
3. Why do participants consuming nuts several times a week or never have higher depression, anxiety, and stress subfactor scores than those consuming daily? This needs to be addressed in the discussion section.
4. Why do participants consuming soybeans several times a week or never have higher anxiety and/or stress subfactor scores than those who consume daily? This needs to be addressed in the discussion section.
5. What were some limitations of the study?
Minor comments:
1. The first paragraph of the introduction has a different space setting than the other parts of the manuscript.
2. The word fructooligosaccharides on Page 2 line 52 has to be changed to Fructooligosaccharides.
3. The number of females described in the results section on Page 5 line 188 has to be changed from 87.5% to 85.5% as shown in Table 1.
4. The median GSRS total scores described in the results section on Page 7 line 215 need to be changed from (U = 13893; p < 0.001) to (U = 18817.5; p < 0.001) as shown in Table 2.
5. The reference to Table 3 on Page 9 line 239 needs to be changed to reference Table 2.
6. The statement that participants who did not consume any probiotic yogurt had depression subfactor scores 6.962 times lower on Page 10 lines 272-273 is incorrect. According to Table 4, it is shown as 6.962 times higher, this needs to be corrected.
7. Why does the statement on Page 11 line 299 have two p-values? The p-values should be added with the number of subfactor scores.
8. The number of pages is restarting after Page 14.
Comments on the Quality of English LanguageThe comments can be found in the section "Comments and Suggestions for Authors
"
Author Response
Thank you for taking your valuable time to evaluate and contribute to our work.

Reviewer 2 Report
Comments and Suggestions for Authors
Peer-review report of the research article (healthcare-2703947)
The manuscript entitled, “Probiotics and Prebiotics Affecting Mental and Gut Health” is an excellent piece of research submitted for publication in the journal “healthcare.”
This manuscript needs a major revision before acceptance.
Probiotics and prebiotics are always a hot topic because of their established implications on human health. Therefore, the idea of the present study is considerably good.
However, the manuscript has several discrepancies that need addressing.
As this article is about gut and mental health, the authors have included questions regarding gut health. Still, questions regarding mental health are missing, such as if the participants are living in a stressful environment or if the people living with them are relaxing or unhealthy people.
The interpretation of results is not appropriate. The authors have discussed the results; however, they have yet to give the reasoning or implications of those results. There is a need to establish reasoning for the gained results. For example, in lines 268-271, the authors demonstrated the results concerning using ayran and yogurt. In the case of ayran, the depression factor is lower, while it is higher in yogurt users at the same frequency. It seems the opposite, but there is no reason for such variation. Such results must be discussed throughout the manuscript.
Moreover, in some places, the authors used the notion "compared to," but didn't present any finding on whether one is better. An example of such a notion is the results given in lines 232-236. Such results must be discussed throughout the manuscript.
The authors have yet to establish the relationship between important factors, such as income and the frequency of probiotic/prebiotic use. It seems logical that people with higher incomes may use more probiotics/prebiotics than those with low incomes. Is there any possible relationship between them?
Line 195. Are they consuming regularly, and how frequently are they using it?
Line 196. Did they not use probiotics, or did they not use any food containing probiotics?
Table 1. Diagnosed Diseases (other than GI) - Which disease is persistent in those people? Is it linked to some psychological condition or not?
Table 1. Why I don't usage probiotics - Is the statement correct, "I do not distrust the content?"
The English language and grammar need improvement.
Author Response

(The authors gave the same response as above.)

Reviewer 3 Report
Comments and Suggestions for Authors
Merve et al. submitted a manuscript entitled: Probiotics and Prebiotics Affecting Mental and Gut Health, in which they mainly talked about the relation between probiotic and prebiotic supplement and gut/mental health with the aid of questionnaire. The authors drew a conclusion that both the consumption of probiotics and prebiotics will be helpful to improve gut health and mental status. Generally, although it is a routine work to prove the benefits of pro(e)biotics, the data seems statistically reliable. This work can be a supplementary evidence to prove the efficacy of pro(e)biotics. It is good that the authors have fully described the methodology, and the cited references are new and related to the topic. However, I think some details may be added to make the manuscript more informative to potential readers.
a) It is strongly recommended for the authors to attach the questionnaire they used into the supplemental data.
b) It might only account for a small amount of participants, but I think it is necessary to exclude the people who were using antibiotics during the survey time. Antibiotic use will strongly interfere with the function of pro(e)biotics.
c) The discussion part can be improved:
1. 2nd paragraph: It is good that the authors listed some literature for comparison. But this part seems lengthy and not organized. In my opinion, to emphasize the key points of this paragraph (I assumed it should be the results in literature and the possible reasons why different conclusions were drawn), the authors can try to reduce the description on methods. Only focus on the differences in these reports to support the assumption that factors such as timing of administration, etc. are important in this kind of reports.
2. 3rd paragraph: The authors started with "the review underscores". It made me confused that "the review" refer to this manuscript or another reference. The authors should use sentence like: Sikorska et al. reported...
3. 4th paragraph: The authors are too arbitrary to say that "the frequency of consumption of bananas, prebiotic food, had a statistically significant effect on the total GSRS scores of the participants." From this work, I cannot tell food consumption like bananas can improve total GSRS scores because banana can function as prebiotics? Or, for example, adjustment diet structure? to improve the mental status.
4. Because so many factors are involved in this questionnaire, I find it difficult to attribute the improvement of gut or mental health to single probiotics or prebiotics use? Or they have syngeneic effect? The authors can talk more about it.
d) Conclusions: Since the conclusion part is the summary of this manuscript. I think the authors can omit the discussion of literature and talk more of the conclusions of this manuscript.
Comments on the Quality of English LanguageLanguage used can be understood.
Author Response

(The authors gave the same response as above.)

Reviewer 4 Report
Comments and Suggestions for Authors
This study revealed the positive relationship between the intake of probiotic and prebiotic foods and mental and gut health. The data were collected through a questionnaire form administered to 538 students enrolled at Afyonkarahisar Health Sciences University. The research in this area is interesting. I only has a suggestion: As the gut microbiota is associated with psychological phenomena such as stress and anxiety by communicating with the brain in two ways. I think the authors should detect the effect of probiotic and prebiotic consumption frequency of participants on stool by using the 16S rDNA sequence.
Author Response

(The authors gave the same response as above.)

Round 2
Reviewer 2 Report
Comments and Suggestions for Authors
The authors have substantially improved the manuscript and have addressed the questions and suggestions. Therefore, the manuscript could be accepted for publication.
Comments on the Quality of English LanguageMinor discrepancies regarding language and grammar need attention, such as in line 99, the authors wrote, "ecluded." According to the context, "excluded" should have been the right word.
Author Response
Dear Reviewer, thank you for your valuable time.
In line 99 "We ecluded the participants who were diagnosed with mental diseases and using any kind of medicine or drug." sentence has been changed to "We excluded the participants who were diagnosed with mental diseases and using any kind of medicine or drug."
